# The Improvement of the Wear Resistance of T15 Laser Clad Coating by the Uniformity of Microstructure

**Yingtao Zhang** [1,*]**, Yongliang Ma** [1]**, Mingming Duan** [1]**, Gang Wang** [2] **and Zhichao Li** [3]

1    College of Mechanical & Electrical Engineering, Hohai University, Changzhou 213022, China
2    Beijing Key Lab of Precision/Ultra-Precision Manufacturing Equipments and Control, Tsinghua University, Beijing 100084, China
3    DANTE Solutions, Inc., Cleveland, OH 44130, USA
*    Correspondence: zhangyt@hhu.edu.cn

**Abstract:** The uniformity of microstructure and wear properties exist in the T15 coating for the laser cladding on 42CrMo steel. It can be improved by a post-heat treatment process. Temperature ranges from 1100 to 1240 °C were applied on the cladding layer to investigate the effect of the heat treatment on the wear resistance and hardness gradient. The post-heat treatment can efficiently improve the inhomogeneity of microstructure. The lower wear rate is obtained after the quenching process at 1100 °C, and the wear rate is increased though the tempering process. The carbides at the grain boundary are decomposed and integrated into the matrix during the high temperature quenching process. The carbides are precipitated and dispersed in the grain during the tempering process. The content of martensite and alloy carbide is significantly increased through the heat treatment process. The microhardness of the cladding layer is 910 HV after quenching and 750 HV after tempering. The wear mechanism of the cladding layer is mainly abrasive wear and fatigue wear. The crack and falling off from cladding layers are significantly reduced for the quenching–tempering process.

**Keywords:** laser cladding; heat treatment; inhomogeneity; T15; wear resistance





## 1. Introduction

42CrMo steel was widely used in the manufacturing of driving gear rings for tracked vehicles. Due to the high load and poor working conditions, the higher hardness and strong wear resistance was required for the part's surface [1]. As an advanced coating preparation and surface modification technology, laser cladding provided an effective solution for the multifunctional coating with higher hardness and better wear resistance for the common alloy steel [2]. The fine grain structure, small heat affected zone and matrix deformation could be achieved by using a laser cladding process [3]. The life and surface properties were promoted by the laser cladding process with relatively small consumption of high-performance materials [4–7].

The wear rate could be decreased remarkably by cladding different materials with high strength and wear resistance [8–10]. The surface wear resistance was 2.4 times the original 42CrMo steel by applying the NiCrBSi/Mo composite coating for the roller [11]. The weight loss of NiCr-TiC composite cladding layer is 1/3 of the stainless steel substrate [12]. As a high-speed steel, T15 is applied broadly in cutting tools, stamping dies and other manufacturing fields, because of its high strength, high hardness and excellent wear resistance [13–15].

As defects formed in the coating layer during the laser cladding process, the optimization of process parameters and material alloy composition was the key study point [16]. The average grain size and porosity of the cladding layer are reduced by ultrasonic-assisted vibration [17]. The cracking phenomenon of T-800 coating improved laser cladding assisted with pre-heat [18]. The Ni60 laser cladding layer with refined grains and no crack can be obtained by applying CeO2 and Y2O3 [19].

The distribution of alloying elements was inhomogeneous in the coating under the rapid heating and cooling rate [20,21]. The inhomogeneity distribution of alloy composition and microstructure had great influence on the coating hardness and wear resistance [22–26].

The microstructures of the heat-affected zone (HAZ) and properties of the cladding layer could be improved by the post-heat treatment [27]. The phase proportion of FeCoCr-NiAlx laser-cladding laser layer could be controlled by using auxiliary heat treatment, resulting in the effective improvement for the mechanical properties [28].

In this paper, the inhomogeneity of T15 coating on 42CrMo steel was studied and the post-heat treatment was put forward to improve the microstructure uniformity and tribological properties of the cladding layer. The microstructure, phase, hardness and tribological properties were analyzed for the original and heat-treated coatings. The wear mechanisms of the coatings were also investigated.

## 2. Experimental Methods

42CrMo steel was used as the substrate with the dimension of $100 \times 100 \times 10$ mm. Before the laser cladding experiment, the oxide layer and oil stain on the surface should be removed. T15 powder with the particle size of 50~80 μm was selected as the coating matrix powder, as shown in Figure 1. The powder was heated in a vacuum drying box at 90 °C for 2 h. The chemical compositions were obtained by a direct-reading spectrometer for 42CrMo and by ICP-OES for the T15 powder, and the testing results are shown in Table 1.

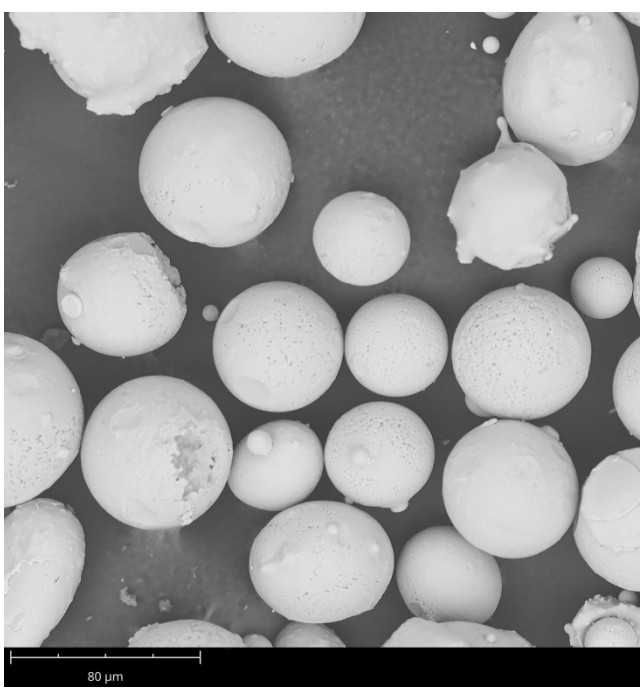

**Figure 1.** Micromorphology of T15 powder.

**Table 1.** Chemical composition of 42CrMo and T15 powder (wt. %).

|  | C | Co | Cr | Mo | Mn | Si | W | V | Fe |
|---|---|---|---|---|---|---|---|---|---|
| 42CrMo | 0.42 | . . . | 0.99 | 0.19 | 0.63 | 0.21 | . . . | . . . | Bal. |
| T15 | 1.6 | 5.4 | 4.5 | . . . | 0.45 | 0.48 | 11.7 | 4.7 | Bal. |

The laser cladding experimental equipment was a Disk laser (TruDisk 4002, Trumpf, Ditzingen, Germany) and a KUKA robot system (KR60-3, KUKA, Augsburg, Germany). Argon gas was used as the powder carrier and protective gas to prevent the melt pool from oxidizing. The protective gas flow was 20 L/min [29]. T15 cladding layer thickness was

about 1 mm. The process parameters of laser cladding are shown in the Table 2. After cladding, the coatings were cut into small specimens with size $10 \times 10 \times 10$ mm by wire cutting for heat treatment and further analysis.

**Table 2.** Laser cladding process parameters.

| Order | Power (W) | Powder Feeding Voltage (V) | Scanning Speed (mm/s) | Overlap Rate |
|-------|-----------|----------------------------|------------------------|--------------|
| 1 | 2000 | 50 | 7 | 30% |
| 2 | 2300 | 40 | 6 | 40% |
| 3 | 2300 | 50 | 8 | 40% |
| 4 | 2300 | 60 | 9 | 40% |
| 5 | 2300 | 70 | 6 | 40% |

The cladding samples were heated by muffle furnace (FB, IRM, Lilienthal, Germany). The cladding layer was quenched at 1100 °C, 1190 °C and 1240 °C, respectively, to study its effect on the microstructure uniformity and wear resistance of the cladding layer. The process parameters of heat treatment were shown Figure 2.

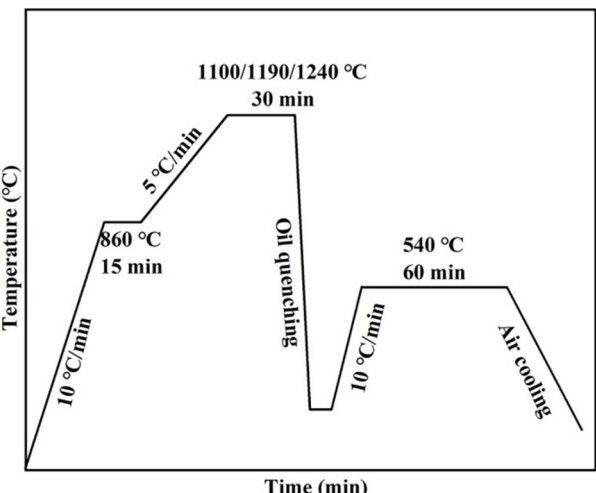

**Figure 2.** Heat treatment process curve of cladding layer.

The hardness of samples was tested by the Vickers hardness tester (HVS-1000A, Huayin, Laizhou, China). The test load was 300 g and the loading time was 15 s. Twenty points with an interval of 0.1 mm were measured continuously, and the average value of three times was taken as the hardness value of this point.

The dry sliding wear properties at room temperature were tested by a CFT-I friction and wear testing system (CFT-I, Zhongke Kaihua, Lanzhou, China). The test configuration is schematically illustrated in Figure 3. The grinding ball was YG6 with a hardness of 93 HRC and a diameter of 6 mm. The wear mark is located in the middle of the non-overlapping area. The test conditions are shown in Table 3.

**Table 3.** Details of the wear test conditions.

| Load (N) | Speed (r/min) | Test Duration (min) | Sliding Distance (m) | Reciprocating Length (mm) |
|----------|---------------|---------------------|----------------------|---------------------------|
| 50 | 100 | 120 | 120 | 5 |

The wear volume ($V$) was measured by the contour scanning instrument. The wear volumes were calculated by using the equation: where $V$ was the volume loss (mm$^3$),

*L* is the experimental load (N) and *S* is total test distance (m). The test configuration is schematically illustrated in Figure 4.

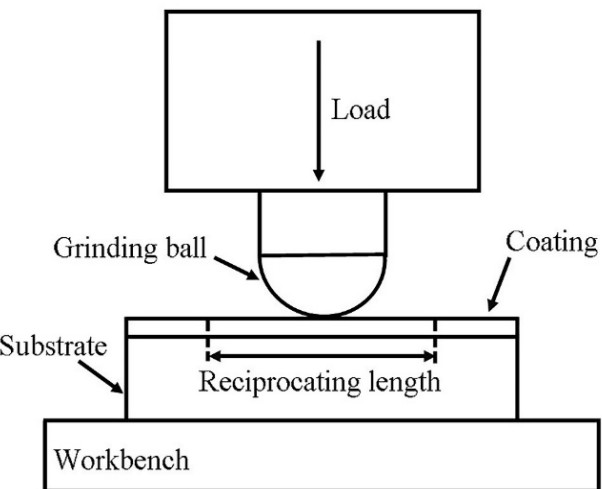

**Figure 3.** Schematic diagram of the wear test.

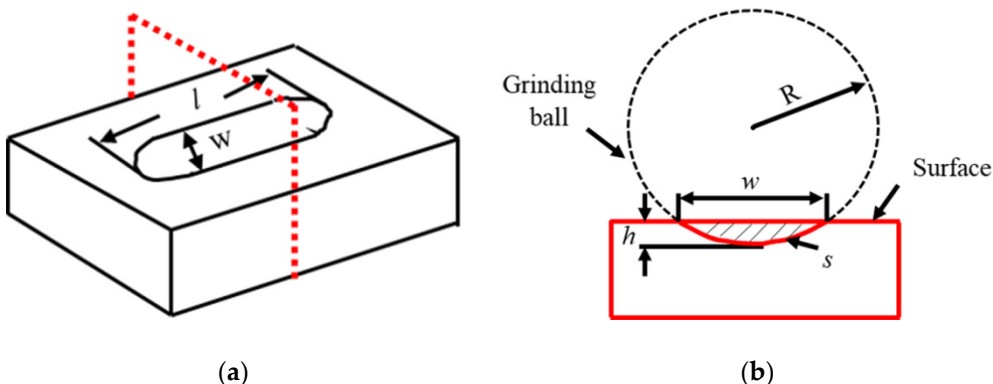

| (**a**) | (**b**) |
|---|---|

**Figure 4.** Schematic diagram of the wear mark: (**a**) sample wear scar, (**b**) wear mark section. (*l* is the length of wear scar, *w* is the width of wear scar, *h* is the depth of wear scar, *s* is the cross sectional area of wear scar, *R* is the Grinding ball radius).

From the calculated wear volume, the wear rates were evaluated using the following Equation (1) [30].

$$W = \frac{V}{L \times S} \tag{1}$$

The microstructures and morphology of the wear trace were characterized by the scanning electron microscope (SEM, ZEISS, Sigma500, Oberkochen, Germany) equipped with energy dispersive spectroscopy (EDS) analysis system. The phase of the coating was identified by X-ray diffraction (XRD, Empyrean, Panaco, Almelo, Holland).

## 3. Results and Discussion

### 3.1. Microstructures and Hardness

After grinding, polishing and corrosion, there were obvious "black-and-white" areas on the surface of all cladding layers. The macro morphology of sample 1 was shown in Figure 5a. The "black-and-white" areas were marked as position 1 and 2, respectively. It can be found from Figure 5b that there was a large hardness gradient on the coating surface. Figure 5c,d showed microstructure of the T15 coating at positions 1 and 2. The coating was mainly composed of the equiaxed crystal and alloy carbides. The average size of grain size at position 2 was about 3.8 µm, which was much smaller than at position 1 with about 7 µm.

The network alloy carbides were situated on the grain boundary, because alloy elements were excluded to the grain boundary in the process of forming equiaxed crystals [31]. The number of stress corrosion cracks at position 1 were more than position 2, which were formed at the combined action of large residual tensile stress and acid corrosive agent [32].

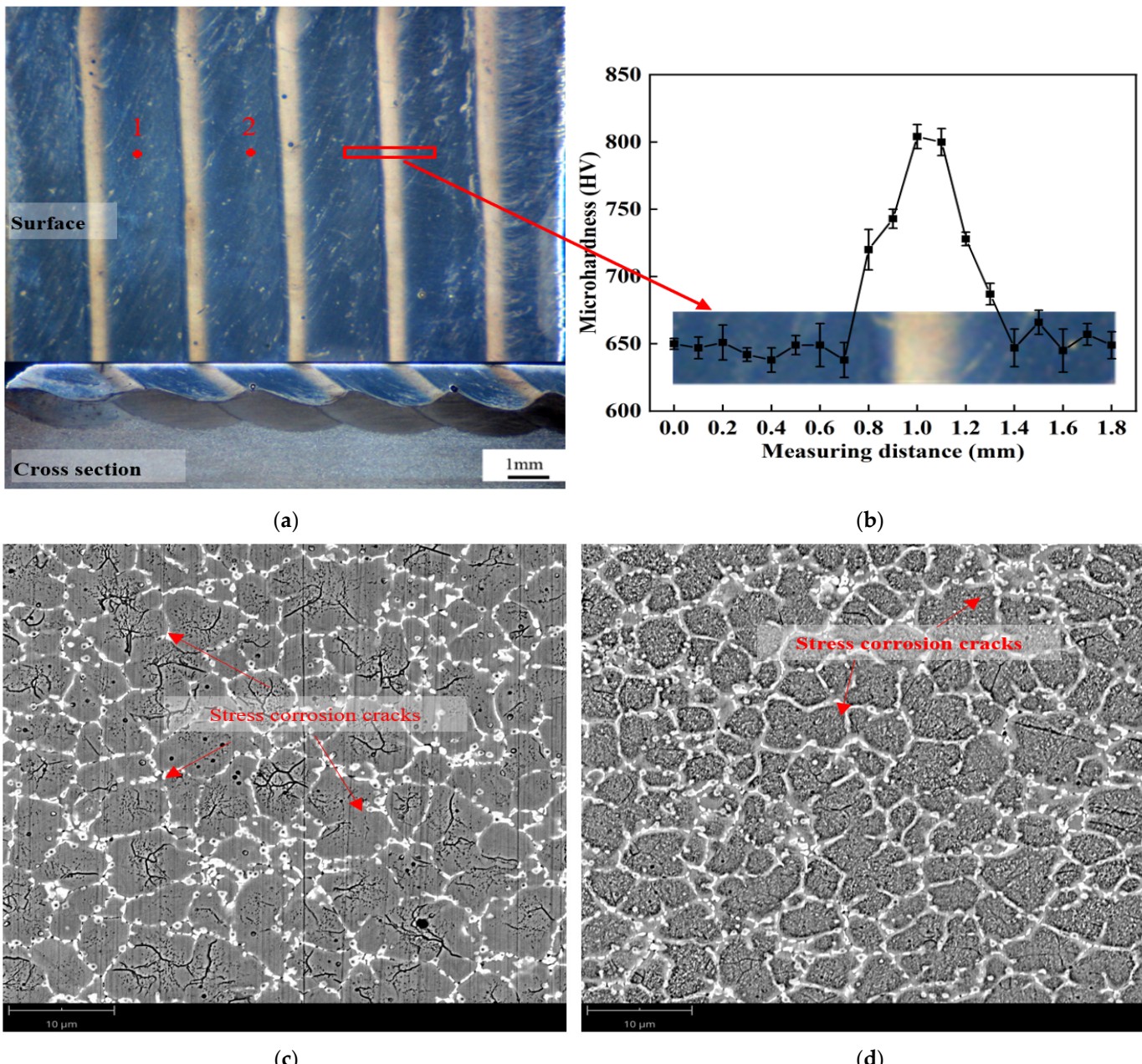

**Figure 5.** Surface macro morphology, microstructure and hardness of cladding layer: (**a**) macro morphology, (**b**) hardness gradient, (**c**) position 1, (**d**) position 2.

Figure 6 shows the surface scanning results of EDS at positions 1 and 2, respectively. Element segregation existed in both "black-and-white" areas of the cladding layer. A large number of W, V and Cr elements were enriched at the grain boundary [33], because these alloy elements with larger atomic radius were difficult to be a solid soluble with iron element [31]. However, the Co element was companied with the Fe element distributed in the inner grain. The lattice structure, atomic radius and electronic structure of the Co element may be similar to that of the Fe element in the matrix [34].

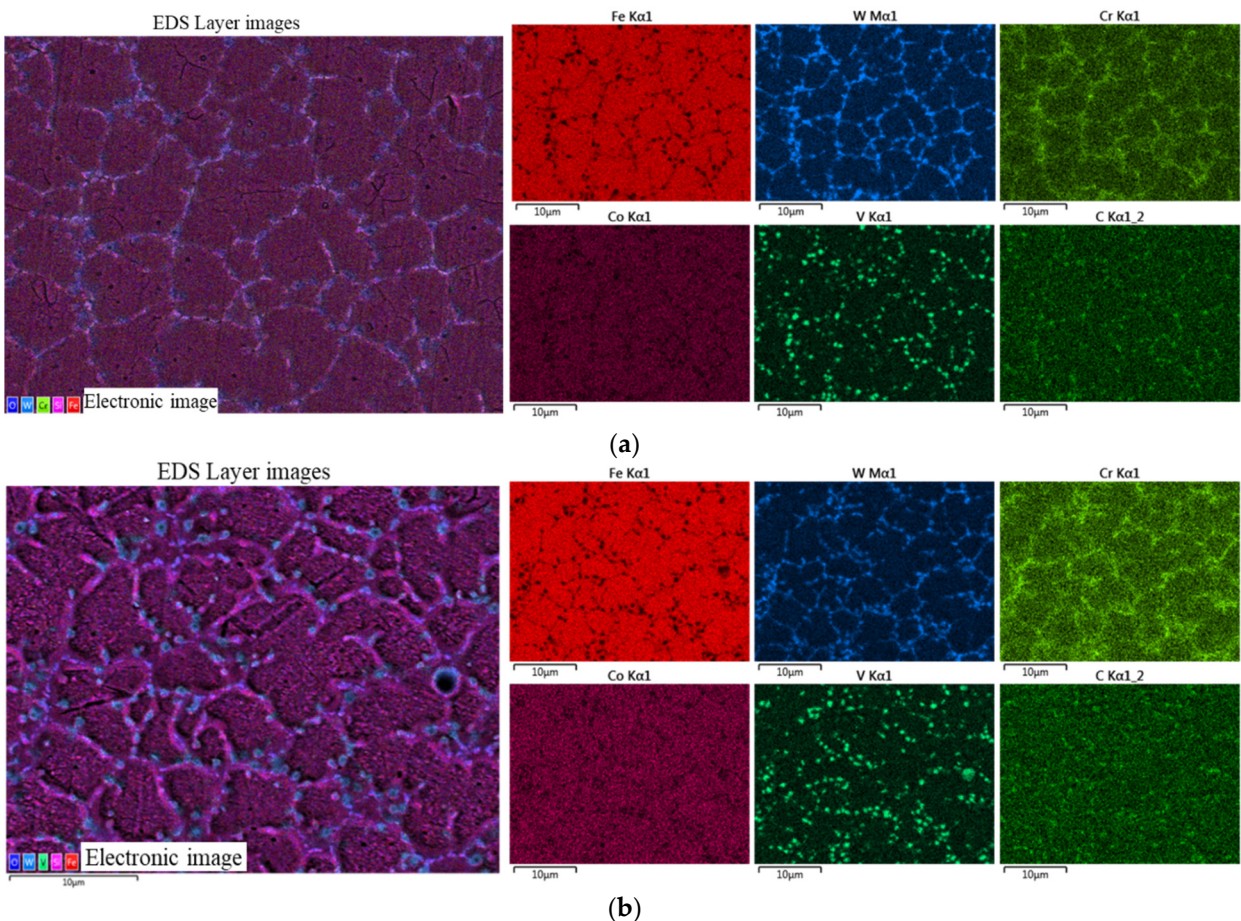

**Figure 6.** The EDS mapping results of T15 coating at different areas: (**a**) position 1, (**b**) position 2.

### 3.2. SEM Morphology after Quenching

A multi-channel transverse lap model was constructed, as shown in Figure 7. A heat-affected zone (HAZ) was only formed on the 42CrMo matrix during the first cladding surface. In subsequent cladding, the number of HAZ was changed because of the remelting zone (RZ). One HAZ was located on the arc surface of the prior cladding layer and the other was located on the 42CrMo substrate. Combined with the macro morphology of the coating, it was speculated that the black area was cladding zone (CZ) and the white bright zone was the HAZ of the arc surface of the front cladding layer.

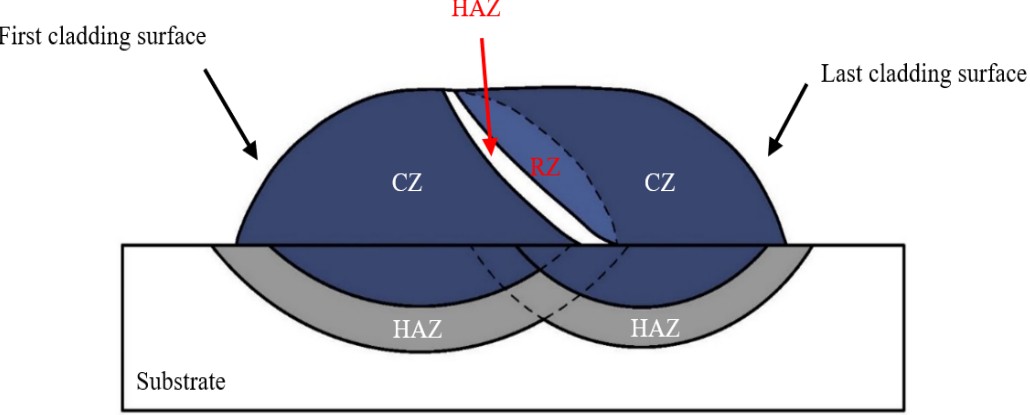

**Figure 7.** Schematic diagram of multi-channel transverse overlapping.

Figure 8 shows the surface macro morphology of sample 1 after different quenching temperatures. The "black-and-white" area of the coatings surface and cross-section was eliminated after quenching. The structural uniformity of the coating was improved because the alloy carbides and elements were melted into the matrix and uniformly precipitated again at high temperature quenching [35].

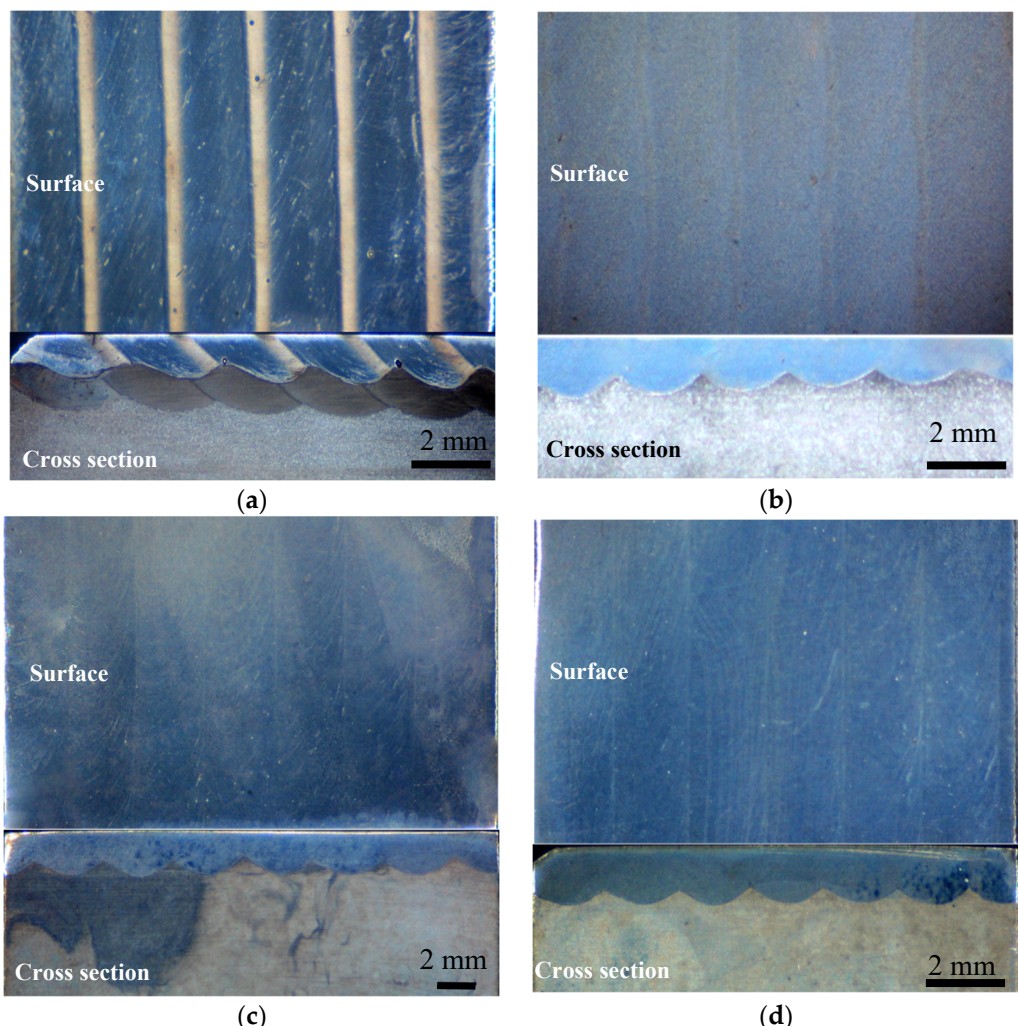

**Figure 8.** Surface macro morphology of cladding layer at different quenching temperatures: (**a**) original coating, (**b**) 1100 °C, (**c**) 1190 °C, (**d**) 1240 °C.

According to the wear rate of sample 1 at different quenching temperatures (shown in Table 4), it can be clearly seen that the wear resistance of the coating can be further improved after quenching at 1100 °C. Therefore, the following focused on the changes of microstructure, hardness and tribological performance of cladding layers at 1100 °C quenching.

**Table 4.** Average wear rate of coatings at different heat treatment processes.

| Heat Treatment Process | Average Wear Rate ($\times 10^{-6}$ mm$^3 \cdot$N$^{-1} \cdot$m$^{-1}$) |
|---|---|
| Original coating | 0.73 |
| Quenched (1100 °C) | 0.54 |
| Quenched (1190 °C) | 1.2 |
| Quenched (1240 °C) | 1.34 |

For the quenched coating (1100 °C), there was no significant difference in microstructure at scanning electron microscope. The surface microstructure of the cladding layer was shown as in Figure 9a. During the process of high temperature quenching, the network alloy carbides were gradually dissolved on the grain boundary of the cladding layer. A large number of fine carbides gradually integrated into the matrix. Some unmelted particles may be VC because of their high melting point [36].

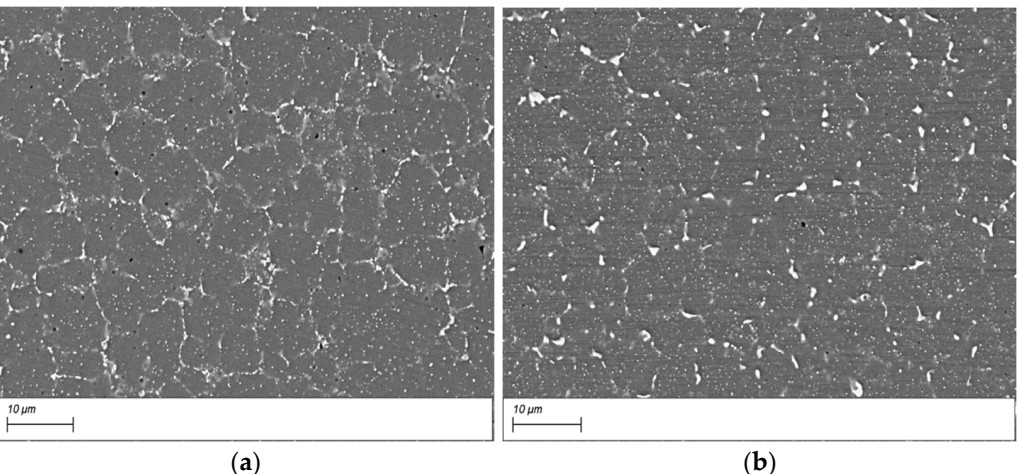

**Figure 9.** The microstructure of the cladding layer after heat treatment: (**a**) quenched, (**b**) quenched-tempered.

After quenching, there were a lot of residual stress, retained austenite and brittle phases in the cladding layer. Tempering was immediately carried out to avoid cracks. Figure 9b shows the microstructure of the quenched cladding layer after tempering treatment. A large number of carbides in the grains of the cladding layer were uniformly distributed on the matrix, which were mainly spherical and very small in size.

As shown in Figure 10a,b, the Fe, W, Cr, and Co alloying elements were uniformly distributed except for the V element after post-heat treatment, which was consistent with the above guess of class V carbides.

Figure 11 shows the XRD patterns of the original and heat-treated T15 cladding layer. The phases of original CZ and HAZ could not be measured, respectively, due to the large diameter of diffraction spot. Therefore, the phases of two different regions were not distinguished. The phases compositions of the coatings after heat treatment were similar to those of the original coating, which mainly consisted of martensite, austenite, MC and M6C carbide. The percentage of alloy carbide increased obviously after the quenching and tempering process, for the reason of the secondary precipitation of carbide during heat treatment. Compared with the original cladding layer, the martensite diffraction peak of heat-treated cladding shifted to the right and changed from double-peak to single-peak, which indicated that the martensite structure had been changed by the heat treatment.

As seen from Figure 12, the hardness gradient of the cladding layer could significantly be eliminated after heat treatment. The cladding layer increased to 910 HV after quenching. This is because a large amount of martensite was formed in the cladding layer. After quenching–tempering, the hardness reduced to 750 HV, because of the decomposition of martensite or without good secondary hardening.

### 3.3. Wear Rate and Friction Coefficient

As seen from Figure 13, the cladding layer had experienced running in period and stable period during the wear process. The friction coefficients increased rapidly during the running-in period, which was accompanied with increased wear rate of T15 coatings. The friction coefficient tended to be stable from rising, falling and rising. The friction coefficients of the coatings varied from 0.5 to 0.8.

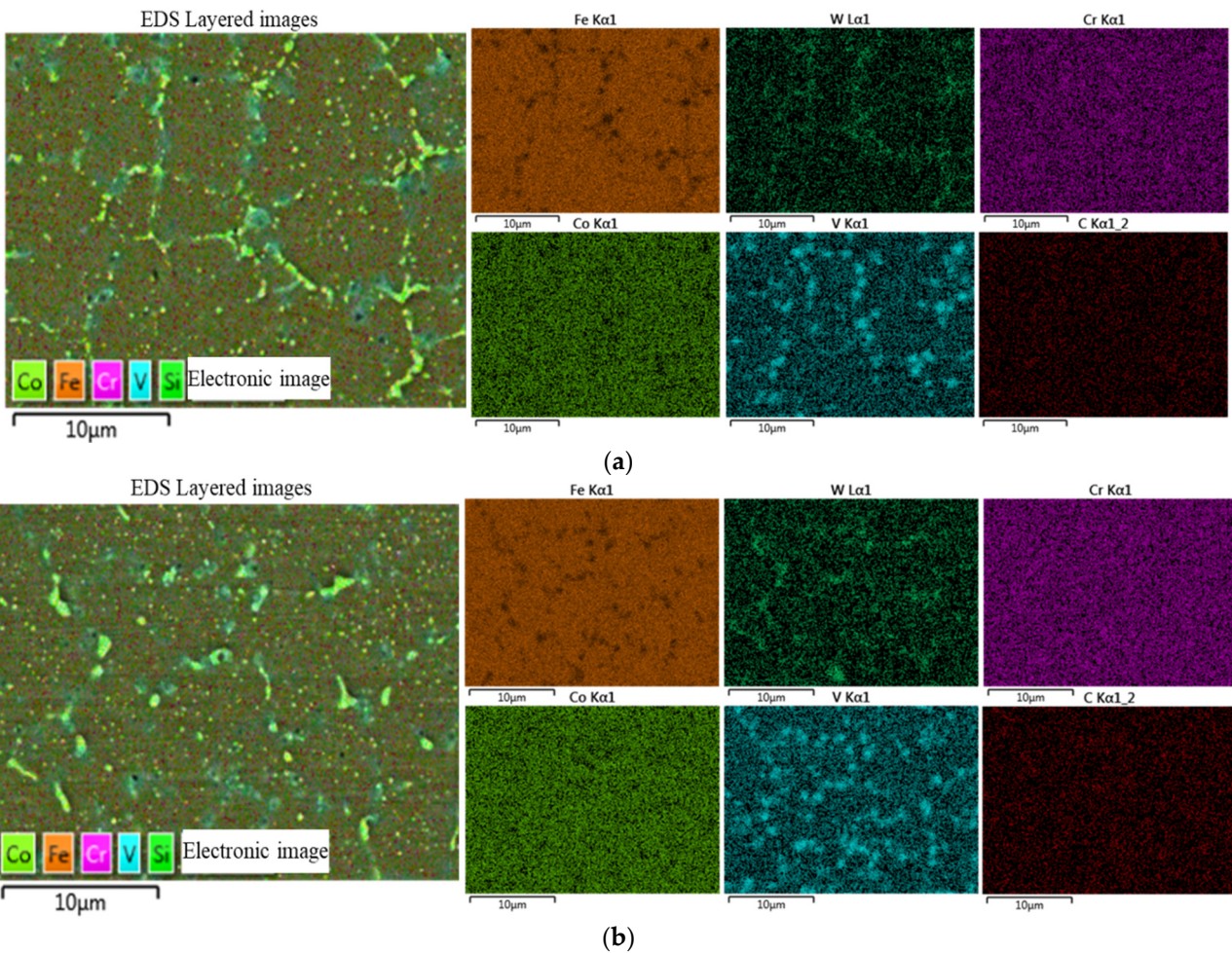

**Figure 10.** EDS area scanning results of the cladding layer after heat treatment: (**a**) quenched, (**b**) quenched–tempered.

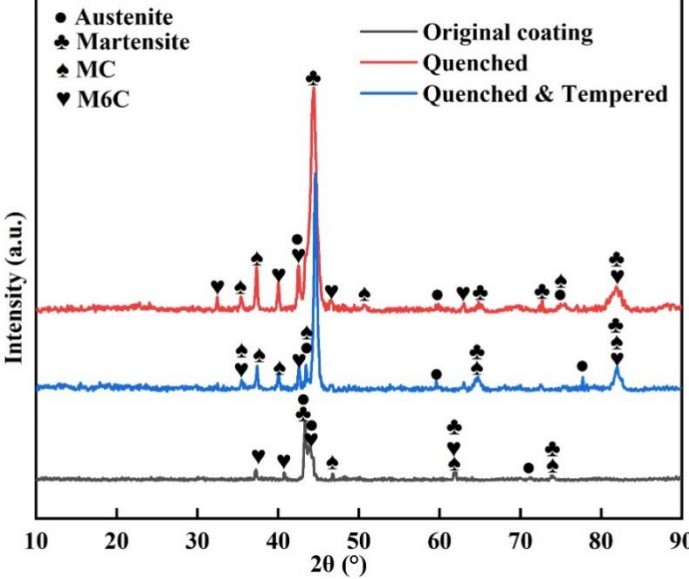

**Figure 11.** XRD patterns of the T15 coating.

For original CZ, HAZ, quenched and quenched and tempered cladding layers, the average friction coefficients were 0.7, 0.71, 0.76 and 0.68, respectively, as shown in Figure 14.

Post-heat treatment had no significant effect on the friction coefficient, because it was a comprehensive characteristic coefficient affected by many factors. The small friction coefficient of tempered samples did not mean that its wear rate was small [37]. The average wear rate of original CZ ($7.3 \times 10^{-7}$ mm$^3 \times$ N$^{-1} \times$ m$^{-1}$) was nearly twice that of HAZ. After the quenching and quenching–tempering process, the microstructures and hardness gradient of the cladding layer was uniform. The wear rate was $5.6 \times 10^{-7}$ mm$^3 \times$ N$^{-1} \times$ m$^{-1}$ and $8.5 \times 10^{-7}$ mm$^3 \times$ N$^{-1} \times$ m$^{-1}$ respectively.

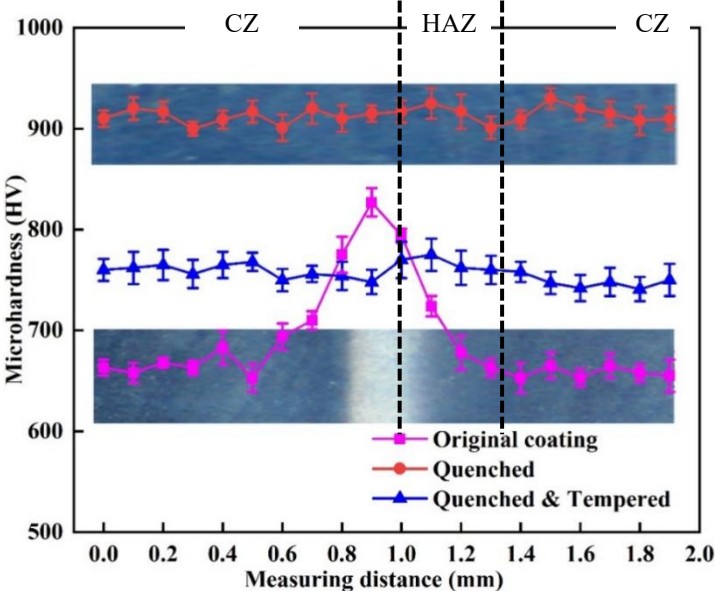

**Figure 12.** Surface microhardness of cladding layer.

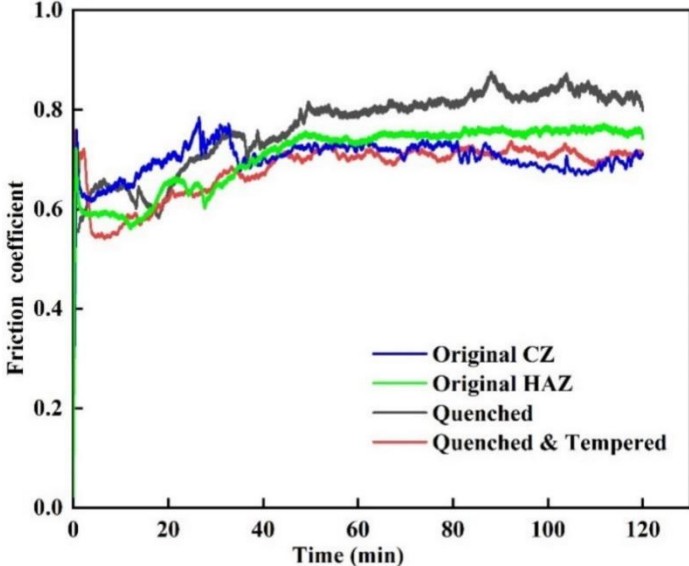

**Figure 13.** Friction coefficient of cladding layer.

Figure 15 showed the wear trace morphology of different cladding layers. As shown in Figure 15a, there were a large number of fatigue cracks, spalling and furrows on the wear surface of original CZ for large residual stress and brittle phases. It indicated that the main wear mechanism was fatigue failure and abrasive wear [38]. Under the cyclic alternating load, microcracks appeared on surface, and then the residual stress promoted the propagation of microcracks into macrocracks [39]. After quenching and quenching–tempering, the wear mark surface of the sample was relatively smooth. This was due to

the hardness improving and the residual stress reducing, avoiding the initiation and rapid propagation of microcracks [40]. In addition, the number of fatigue cracks and the fatigue shedding phenomenon was significantly reduced, as shown in Figure 15b,c. The results showed that the wear mechanism was still mainly slight-abrasive wear and fatigue wear.

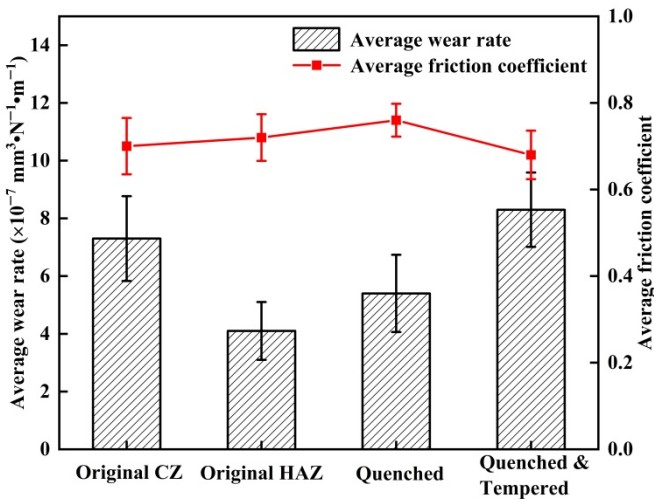

**Figure 14.** Average wear rate and friction coefficient of cladding layers.

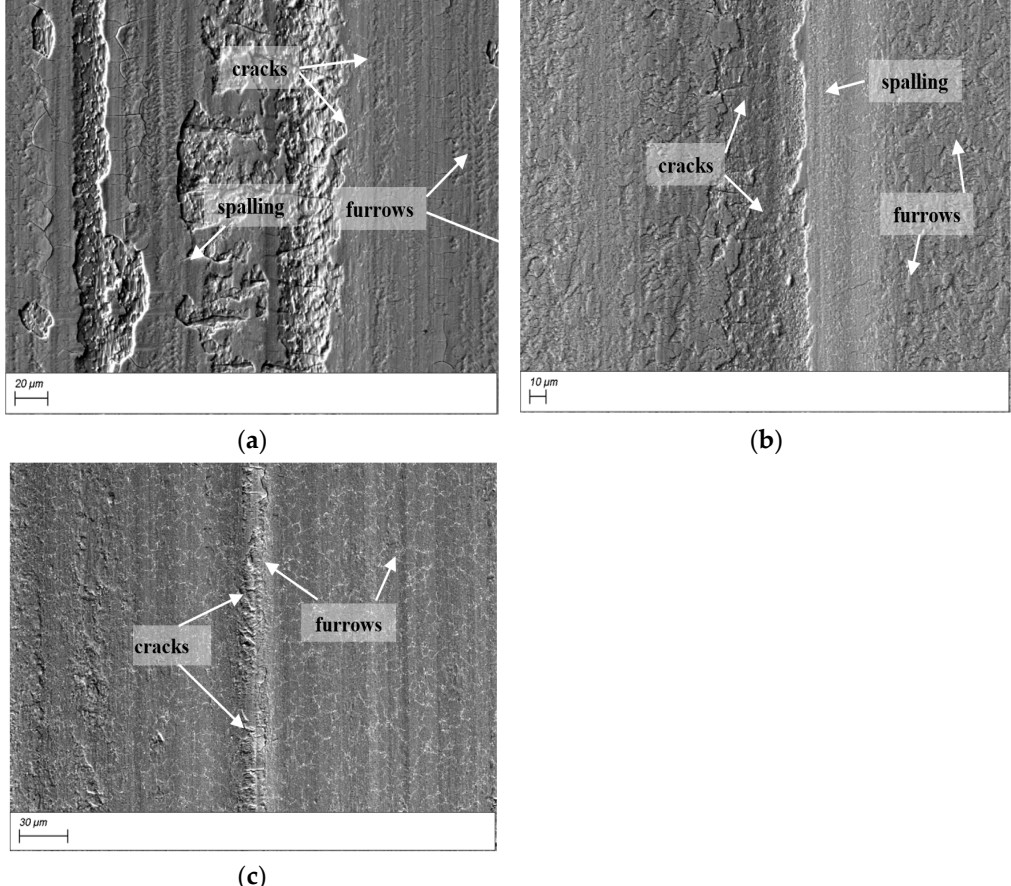

**Figure 15.** The SEM wear trace morphology of cladding layer: (**a**) original CZ, (**b**) quenched, (**c**) quenched–tempered.

Compared with quenched and tempered samples, there were many fatigue cracks in the wear marks of quenched samples, because there were a lot of brittle martensite

and residual stress on the surface. The plasticity and strength of tempered coating was improved for a lot of dispersed carbides. Due to the low surface hardness of the tempered sample, deep furrows were formed under the action of hard wear debris.

As seen from Figure 16, the average friction coefficient of coatings with different parameters hovered around 0.7. For different coatings, the standard deviation of the average wear rate was 0.21 mm$^3 \times$ N$^{-1} \times$ m$^{-1}$. After quenching (1100 °C) and quenching–tempering (1100 °C–540 °C), it was reduced to 0.07 and 0.09 mm$^3 \times$ N$^{-1} \times$ m$^{-1}$, respectively. According to the wear data of the above coating with different parameters, the experimental accuracy was verified by the same post-heat treatment process.

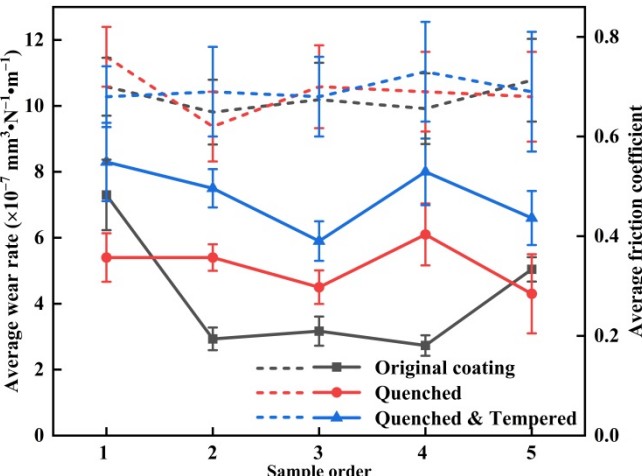

**Figure 16.** Average wear rate and friction coefficient of coatings with different parameters.

## 4. Conclusions

There were obvious "black-and-white" areas (cladding zone and heat-affected zone) on the surface of the T15 cladding layer. The average size of equiaxed grains was finer. The microhardness was about 200 HV higher in the CZ than HAZ. The average wear rate of original CZ and HAZ was $7.3 \times 10^{-7}$ mm$^3 \cdot$N$^{-1} \cdot$m$^{-1}$ and $4.3 \times 10^{-7}$ mm$^3 \cdot$N$^{-1} \cdot$m$^{-1}$. The alloy compounds were distributed in the grain boundaries for both regions. The microstructure uniformity and hardness gradient of the surface was eliminated by the post-heat treatment. The alloy elements except for V were dissolved into grains during the quenching process.

The microhardness of cladding layers was 910 HV after quenching and 750 HV after tempering. Post-heat treatment had no significant effect on the friction coefficient. After quenching and quenching–tempering, the wear rate of sample 1 was $5.6 \times 10^{-7}$ mm$^3 \cdot$N$^{-1} \cdot$m$^{-1}$ and $8.5 \times 10^{-7}$ mm$^3 \cdot$N$^{-1} \cdot$m$^{-1}$, respectively. The wear mechanism of the cladding layer was mainly abrasive wear and fatigue wear. The quenching residual stress could be eliminated by the subsequent tempering process.

For the cladding layers with different cladding process parameters, the average friction coefficient of coatings hovered around 0.7. In addition, the standard deviation of wear rate was decreased after post-heat treatment.

**Author Contributions:** All of the authors contributed to this idea and method. Sample preparation, data acquisition, and analysis were all performed by Y.M. The first draft of the manuscript was written by Y.M. Corresponding author, Y.Z., read the comments on the previous version of the manuscript and approved the revised manuscript. All authors have read and agreed to the published version of the manuscript.

**Funding:** This work was financially supported by the National Natural Science Foundation of China (No. 51905148 and No. 51875169) and by the Fundamental Research Funds for the Central Universities (No. B200202218).

**Data Availability Statement:** All data that support the findings of this study are available from the corresponding author upon reasonable request.

**Conflicts of Interest:** The authors declare no conflict of interest. The funders had no role in the design of the study, in the collection, analyses, or interpretation of data, in the writing of the manuscript, or in the decision to publish the results.

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
