# Peer review of "The Improvement of the Wear Resistance of T15 Laser Clad Coating by the Uniformity of Microstructure"

_lubricants, doi:10.3390/lubricants10100271_

Round 1
Reviewer 1 Report
The author has done a lot of work to investigate the improvement on the uniformity of microstructure and wear properties for T15 laser clad coating. This work has a certain application prospect, but the work content and structure need to be further improved.
1.There is only a subheading 3.1 under the section of “3.Results and discussion”, but “3.2” or “3.3” can not be found, which need to be improved.
2. Descriptions should be done in conjunction with Figure. For example, line 244-248.
3. The mechanism of wear resistance and the friction coefficient should be explained clearly.
4. The authors discussed the “average wear rate and friction coefficient” of sample 1-5, however, this is not relevant to the topic of this article.
5. There are a lot of editing and grammar errors. Such as multiple punctuation and capitalized first letter problems “The wear volume (V) was measured by the contour scanning instrument. the wear volumes were calculated by using the equation: , where V is the volume loss (mm3), L is the experimental load (N) and S is total test distance (m).”, grammar errors “The average size of equiaxed grains was finer and the microhardness was higher about 200HV in the CZ than HAZ.”
Author Response
Dear Reviewer:
We are very grateful to your careful review and constructive suggestions. The comments are all valuable and helpful for revising and improving our paper. Below is our response to your comments.
Comment 1. There is only a subheading 3.1 under the section of “3.Results and discussion”, but “3.2” or “3.3” can not be found, which need to be improved.
Reply: Thanks for the reminding. The subheading 3.2 and 3.3 was missing in our submitted manuscript. We've added “3.2 SEM morphology after quenching, 3.3 wear rate and friction coefficient” in the manuscript.
Comment 2. Descriptions should be done in conjunction with Figure. For example, line 244-248.
Reply: As suggested by your comment, all descriptions of Figures were checked and the relationships between them were improved.
Comment 3. The mechanism of wear resistance and the friction coefficient should be explained clearly.
Reply: Considering your comment, The mechanism of wear resistance and the friction coefficient for different samples were revised. And the more explanations were added in the corresponding part.
Comment 4. The authors discussed the “average wear rate and friction coefficient” of sample 1-5, however, this is not relevant to the topic of this article.
Reply: We were sorry for that this part might not be explained clearly. In this paper, the effect of post heat treatment on the laser cladding layer was studied. The inhomogeneity of microstructures, hardness and wear resistance was reduced by the heat treatment process. The “average wear rate and friction coefficient” are the main components of wear resitance. So the discussions about the “average wear rate and friction coefficient” were closely related to the topic of this article..
Comment 5. There are a lot of editing and grammar errors. Such as multiple punctuation and capitalized first letter problems “The wear volume (V) was measured by the contour scanning instrument. the wear volumes were calculated by using the equation: , where V is the volume loss (mm3), L is the experimental load (N) and S is total test distance (m).”, grammar errors “The average size of equiaxed grains was finer and the microhardness was higher about 200HV in the CZ than HAZ.”
Reply: Thanks for your reminding of editing and grammar errors. We checked about these errors and revised them through the manuscript three times.

Reviewer 2 Report
Dear Authors,
in the manuscript titled: "The improvement of the wear resistance of T15 laser clad coating by the uniformity of microstructure" you write about the possibility of improving the functional properties by increasing the durability of the friction elements of the drive transmission. The work is interesting and the presented results are new and interesting. Generally, the work is written in correct language with a fairly well-described methodology and presentation of results and discussion. The conclusions are quite general but accurate. Unfortunately, there are some shortcomings in the work that need to be corrected and which I indicate below:
1. Please align the text of the abstract.
2. The authors chose the literature properly, but they limited themselves very much geographically, so I propose to extend it with the following publications:
https://doi.org/10.3390/ma14020239
https://doi.org/10.1007/s00170-016-8704-3
https://doi.org/10.3390/met10070856
https://doi.org/10.3390/ma13204687 can be inserted in lines 37-39.
3. Change the font in the Tables: 1, 2, 3, 4.
4. The text in lines 97-107 is unclear and should be corrected.
5. I don't know if chapter 3.1 is needed.
6. Figures 6 and 10 should be enlarged as it is illegible.
7. Figure 7 may be slightly smaller.
8. Change the format of captions under the drawings - it will match them to the journal template.
9. Similarly, match the reference catalog to the template.
Thank you.
Author Response
Dear Reviewer:
Thank you for your careful review constructive suggestions. We would like to express our great appreciation for your comments about the format problems for the manuscript. Based on these comments and suggestions, we have made careful modifications on the original manuscript. We hope the revised manuscript will meet with the approval. Below is our point-by-point response to your comments.
Comment 1. Please align the text of the abstract.
Reply: The text of the abstract was aligned.
Comment 2. The authors chose the literature properly, but they limited themselves very much geographically, so I propose to extend it with the following publications:
https://doi.org/10.3390/ma14020239
https://doi.org/10.1007/s00170-016-8704-3
https://doi.org/10.3390/met10070856
https://doi.org/10.3390/ma13204687 can be inserted in lines 37-39.
Reply: All references that you suggested were added to the appropriate location.
Comment 3. Change the font in the Tables: 1, 2, 3, 4.
Reply: The size and format in Tables were revised for font.
Comment 4. The text in lines 97-107 is unclear and should be corrected.
Reply: As you suggested, the text was revised to make the explanations more clearly.
Comment 5. I don't know if chapter 3.1 is needed.
Reply: Thanks for the reminding. The subheading 3.2 and 3.3 was missing in our submitted manuscript. We've added “3.2 SEM morphology after quenching, 3.3 wear rate and friction coefficient” in the manuscript.
Comment 6. Figures 6 and 10 should be enlarged as it is illegible.
Reply: Figures 6 and 10 were the SEM observations of cladding layer, proving the improvement of heat treatment on the microstructure uniformity. The difference of microstructures and alloying compound distribution around grain boundary was clear. The clear original pictures were uploaded to the submit system. The size of the Figures could be determined by the editor.
Comment 7. Figure 7 may be slightly smaller.
Reply: Thank for you recommendation and the size of Figure 7 was changed from 145*60 mm to 105*44.
Comment 8. Change the format of captions under the drawings - it will match them to the journal template.
Reply: According to your suggestions, the format of captions under the drawings was totally checked and revised to the journal template.
Comment 9. Similarly, match the reference catalog to the template.
Reply: Format of References was carefully checked and modified one by one to match the template.

Reviewer 3 Report
Dear authors, you have presented an interesting idea to modify the surface of 42CrMo steel. T15 high speed steel in terms of wear resistance is a good choice as it has excellent resistance in anti-wear applications. Please reflect on my comments and suggestions for research:
- The introduction is quite general, there are too few references.
- Regarding the heat treatment of T15 steel, the dissolution of M6C carbides starts above about 1100 degrees Celsius, while one-time tempering is a mistake, at least 2 are required for high-speed steels to get rid of the residual austenite completely. And in the drawing from XRD, you can see the austenite peak after tempering and we have a catastrophe ready, it is enough for the working temperature of such a layer to rise and this austenite will change - which will cause cracking.
- Is the tribological test based on a standard or is it an original design?
- This mesh of M6C carbides along the borders is a very unfavorable phenomenon - research on its elimination must be continued, preferably at the stage of layer application.
- Only M6C carbides we are able to dissolve, MC will remain stable as primary carbides - so M6C dissolution starts above about 1100, one might wonder how to carry out this treatment to completely eliminate them from the border, provided that there are also no MC carbides there.
Please consider my suggestions.
best regards
Author Response
Dear Reviewer:
Thank you for your review comments. In response to your review comments, we have made the following modifications and statements.
Comment 1. The introduction is quite general, there are too few references.
Reply: According to the comment, the content of the introduction was enriched and 8 references were added after searching some relevant documents.
Comment 2. Regarding the heat treatment of T15 steel, the dissolution of M6C carbides starts above about 1100 degrees Celsius, while one-time tempering is a mistake, at least 2 are required for high-speed steels to get rid of the residual austenite completely. And in the drawing from XRD, you can see the austenite peak after tempering and we have a catastrophe ready, it is enough for the working temperature of such a layer to rise and this austenite will change - which will cause cracking.
Reply: In this paper, the study point was focused on the improvement of the uniformity for microstructures and wear resistance. After preliminary experiment, quenching+one time tempering can improve the surface structure inhomogeneity of cladding layer. The residual austenite was ignored.
Thanks for the timely reminding. The experiment of 2 times tempering was prepared and the mechanical property testing was designed for the cladding layer after different heat treatment process.
Comment 3. Is the based on a standard or is it an original design?
Reply: The tribological test was an original design. After the tribological testing standard was read, the experimental scheme was designed based on actual part working conditions.
Comment 4. This mesh of M6C carbides along the borders is a very unfavorable phenomenon - research on its elimination must be continued, preferably at the stage of layer application.
Reply: The carbides at grain boundary will affect the corrosion resistance. It also led to the decrease of interface bonding strength, and the grain boundary becomes the origin of fracture.
The way to eliminate the carbides was studied, and the testing of mechanical property about cladding layer was designed. The research work on this project was still going on.
Comment 5. Only M6C carbides we are able to dissolve, MC will remain stable as primary carbides - so M6C dissolution starts above about 1100, one might wonder how to carry out this treatment to completely eliminate them from the border, provided that there are also no MC carbides there.
Reply: Thank you for recommendation on the carbides revolution during the heat treatment process. It was very helpful for us to focus on the observation of distribution and morphology of carbides. TEM test would be applied in the future research work to study the revolution and distribution of carbides.
Round 2
Reviewer 2 Report
Dear Authors,
thank you for making changes to the manuscript. The authors improved the work in the mentioned points, for which I thank you and accept the answers. I noticed a minor error on the list of references, please change it in the final version.
reference 15 it occurs twice, please change to this: https://doi.org/10.3390/ma13204687
Moreover, I recommend the work for printing.
Thank you.
Author Response
Dear reviewer
Thanks for your comments of printing the research work and the modification of reference No.15. According to the suggestions, the reference No.15 was changed to “https://doi.org/10.3390/ma13204687”.
Kind regards.
Your sincerely,
Yingtao Zhang
e-mail:yingtzhang@foxmail.com
Reviewer 3 Report
Thank you for answering my questions/suggestions. After applying corrections related to reviews, the article looks better. However, it seems to me to be more suited to surface engineering journals
Author Response
Dear reviewer
Thank you for your approval of the modification. The research work was submitted to the special issue(Laser Suface Engineering for Tribology) of Journal(Lubricants), which is in accord with the theme of this special issue for the Journal.
Kind regards.
Your sincerely,
Yingtao Zhang
e-mail:yingtzhang@foxmail.com